# A Study about a New Standardized Method of Home-Based Exercise in Elderly People Aged 65 and Older to Improve Motor Abilities and Well-Being: Feasibility, Functional Abilities and Strength Improvements

**DOI:** 10.3390/geriatrics7060134

**Published:** 2022-11-25

**Authors:** Giovanni Melchiorri, Tamara Triossi, Valerio Viero, Silvia Marroni, Giovanna D’Arcangelo, Virginia Tancredi

**Affiliations:** 1School of Sport and Exercise Sciences, Department of Systems Medicine, Faculty of Medicine and Surgery, University of Rome Tor Vergata, Via Montpellier 1, 00133 Rome, Italy; 2Don Gnocchi Foundation IRCCS, Piazzale Rodolfo Morandi 6, 20121 Milan, Italy; 3School of Sport and Exercise Sciences, Faculty of Medicine and Surgery, University of Rome Tor Vergata, Via Montpellier 1, 00133 Rome, Italy; 4Department of Systems Medicine and Centre of Space BioMedicine, University of Rome Tor Vergata, Via Montpellier 1, 00133 Rome, Italy

**Keywords:** fall prevention, home setting, adherence, safety, activity of daily living

## Abstract

Background: To verify the effects in terms of feasibility, strength and functional abilities of a standardized exercise training method that is partially supported (home training), with the aim of improving motor abilities and well-being. Methods: A total of 67 participants underwent two sessions per week for 12 weeks for the program, based on 8 sequences with specific body part targets, with each sequence made up of 9 exercises. Outcome measures: Recording of training session data, Chair Test, Hand Grip Test, Timed Up-and-Go Test, Stork Balance Test, Sit-and-Reach Test, VAS, Perceived Physical Exertion. Results: In total, 97% of the sample were “adherent” (more than 70% of the prescribed treatments performed). The rate of adverse events was infrequent (only 8). Chair Test +31%, Hand Grip Test +6%, Timed Up-and-Go Test −17%, Stork Balance Test +65%, Sit-and-Reach Test +55%, VAS −34%, Perceived Physical Exertion −69%. Conclusions: Home training has good feasibility (adherence, tolerability, safety) and cost-effectiveness ratio and improves both strength and functional abilities, which, in turns, helps to improve motor abilities and well-being.

## 1. Introduction

Home-based exercise is “an exercise session performed in a home setting using body mass, elastic bands or weights as part of an exercise program designed to increase strength and functional ability” [1].

The use of home-based exercise has recently been popularized among many fields of medicine, as shown by several scientific articles demonstrating its applicability in cases of cardiovascular [2,3], orthopedic [4,5,6], neurological [7,8,9], oncological [10,11,12,13], pulmonary [14,15] and nutritional–metabolic disease [16,17,18]. Furthermore, also through very recent studies [19,20,21], the use of home-based exercise has proven to be very useful for general health purposes and fall prevention in elderly people [1,22,23,24], and in balance performance [25].

Interpersonal and environmental motivators of exercise and physical activity must be considered in the design of a home-based exercise program. The support of therapists and interpersonal relations help to improve the patient’s compliance to the program [11,26]. Home-based exercise can be organized as self-controlled (input evaluation, prescribed program, no therapist’s visits), supported (input evaluation, prescribed program, continued presence of the therapist during training) or partially supported (input evaluation, prescribed program, occasional presence of the therapist).

There is evidence to support exercise-based interventions as a cost-effective treatment [27] but great attention must be paid to both the program description and adherence to the training program.

From a systematic review in 2014 [1], it was shown that home-based exercise can improve both strength and functional ability in older men and women, but it also pointed out some limitations found in the international literature. Specifically, the great diversity in training programs (choice of exercises, volume, intensity, progression criteria and subject’s compliance to the program) can make it difficult to understand the real effects of home-based exercise and compare the effectiveness of the various methods.

The improvement of the level of functional abilities and muscle strength, and the consequent reduction of the risk of falls, contribute to a better quality of life of the elderly [1,2,8,13,15,18]. Home-based exercise requires safety, minimal equipment, a home setting and the possibility to implement intensity [17,28]. Therefore, the aims of our study were: (1) to propose a new method for home-based exercise called home training (HT); (2) to identify the effects in terms of the feasibility (adherence, tolerability, safety) of this new method; (3) to verify the modifications induced by this type of training on muscle strength and functional abilities.

## 2. Materials and Methods

### 2.1. Participants

A total of 69 subjects were involved in the study: 25 male and 44 female. Participants were free volunteers randomly selected through social media advertising and recruited if they met the inclusion criteria. To be included in the study, all subjects had to meet the following criteria. Inclusion criteria: (1) age between 65 and 85 years; (2) not practicing any physical activity in a regular way, and in any case performing less than 30 min per week of physical activity; (3) willingness to undergo training at one’s home twice a week under the supervision of a therapist for a period of three months. Exclusion criteria: (1) being affected by chronic developmental pathologies such as rheumatoid arthritis, chronic obstructive pulmonary disease, cardiovascular disease, Parkinson’s or Alzheimer’s disease, dementia; (2) surgery undergone in the past 12 months; (3) being affected by serious cardiovascular disease; (4) being affected by rheumatic disease in the acute phase; (5) outcomes of cerebro-vascular pathologies; (6) presence of active oncological pathology and/or related therapy in the last 5 years; (7) proven disability.

All patients involved in the study were informed of the methods and aims of the research and read and signed a consent form containing this information. All procedures were carried out in accordance with the Declaration of the World Medical Association and the Declaration of Helsinki guidelines. The research was approved by the Internal Research Board of “Tor Vergata” University of Rome.

After a brief interview with the subjects and their physicians, it was ascertained that in the 6 months preceding the start of the treatment, the subjects of our sample had all been in good health. The physicians also gave specific medical authorization for the training.

### 2.2. Outcome Measures

#### 2.2.1. Strength and Functional Abilities Tests

##### 30-s Chair-Stand Test

The 30-s Chair-Stand Test (CHAIR TEST) is a method to measure lower body strength and is commonly used in older adults [29]. The subject starts in a sitting position on a chair, with the arms crossed on his/her chest. At the starting signal, he/she has to fully stand up and sit down again. The total number of movements completed in 30 s is recorded.

##### Hand Grip Test (HAND GRIP)

Hand grip strength measured by dynamometry is well established as an indicator of muscle status, particularly among older adults; moreover, several studies have demonstrated the concurrent relationship of grip strength with measures of nutritional and general health status [30].

The subject performs the test seated on a chair, with his/her arms along the body, the non-dominant limb elbow flexed at 90° and an isometric manual dynamometer in the hand (Lafayette Instrument, model 01163). He/she is asked to perform the maximum isometric contraction.

##### Timed Up-and-Go Test

Timed Up-and-Go test (TUG) is a valid, reliable test used to assess basic functional mobility. It is a quick test that does not need any special equipment or training [31]. The patient, who sits in a chair with arm rests, is required to stand up, walk a distance of 6 m (3 m forward and 3 m back to the chair) at maximum speed and sit down again. The total time taken to complete the whole test is measured.

##### Sit-and-Reach Test

The Sit-and-Reach Test (SIT REACH) is the most common measure of flexibility of the lower back and hamstring muscles. The subject, who is seated on the ground, resting his/her bare feet flat against the side of a box (sit-and-reach box) with a ruler attached to the top, is asked to lean forward and reach along the ruler as far as possible [32].

##### Standing Stork Balance Test

The Standing Stork Balance Test (STORK) is used to measure the static balance of a subject, who is required to maintain a one-leg standing position on the non-dominant leg [33], with the opposite foot against the inside of the supporting knee and both hands on the hips. The stopwatch is started as the heel is lifted from the floor and the time spent in this position is measured [34].

#### 2.2.2. Well-Being Tests

##### Perceived Physical Exertion

Perceived Physical Exertion (PPE) is used as a measure of perceived physical fatigue during daily life activities [35]. Perceived physical exertion is measured using Borg’s scale (CR10), with 0 standing for “nothing at all”, 3 for “moderate”, 4 for “somewhat hard”, 5 for “hard” and 10 for “very, very hard [36]”.

##### Evaluation of Pain

The Visual Analog Scale (VAS) is a tool for pain level assessment. The VAS has proven to be satisfactory in many fields of health sciences. The VAS scale used includes 10-cm lines anchored at the ends by words that define the bounds of various pain dimensions [37]. The VAS was used for the global evaluation of skeletal muscle pain potentially present in the segment of the population investigated.

### 2.3. Intervention

#### Exercises and Sequences

A total number of 72 exercises were provided, grouped into 8 different sequences of increasing difficulty, from the simplest to the most complex, with each sequence having a training target directed to a specific body part and requiring an ever-increasing physical and motor effort as the participant levels up. Of the 8 total sequences, 3 involved the lower limbs, 2 the core stability, 1 the trunk and 2 the upper limbs. Based on an unpublished preliminary study of ours, the sequences were classified as entry-level, medium-level and high-level according to their overall degree of difficulty. The method used did not require special equipment, but only inexpensive and easily available materials: 1 gymnastics mat, 1 rubber ball and 1 elastic band. The therapists involved in the project had undergone a period of training on the method and application criteria and alternated in following the study participants. The therapists, who were all physiotherapists and/or motor scientists, were responsible for planning and applying the HT method.

All subjects started training with entry-level sequences (sequences 1, 2 and 3). Descriptions of sequences 1, 2 and 3 are provided in Figure 1, Figure 2 and Figure 3.

Figure 1, Sequence 1: (1) squat from a chair, hands on table; (2) chair squat; (3) isometric wall squat; (4) wall squat with ball between lower back and wall; (5) squat; (6) squat with shoulder abduction; (7) candelabrum squat; (8) squat alternating arm moves; (9) y squat.

Figure 2, Sequence 2: (1) chair “spinal wave”; (2) wall “spinal wave”, lower back against the wall; (3) isometric back extension (sitting on a chair, trunk–hip angle at 45°, hands on hips); (4) from previous position 3 to “spinal wave”; (5) back extension (sitting on a chair, hands on hips, from trunk–hip angle at 45° to 90°); (6) alternating shoulder back elevation (stand up position, hip–trunk angle at 45°); (7) double back shoulder extension (stand up position, hip–trunk angle at 45°); (8) shoulder back elevation (stand up position, hip–trunk angle at 45°); (9) dynamic movement from exercise position 7 to 8 and return.

Figure 3, Sequence 3: (1) isometric chair low rowing with elastic band (sitting position); (2) chair dynamic full R.O.M. low row with elastic band (sitting position); (3) chair single alternating low row with elastic band (sitting position); (4) isometric low row (with elastic band in standing position); (5) low rowing (with elastic band in standing position); (6) single arm alternating low row (with elastic band in standing position); (7) “w” low row (with elastic band in standing position); (8) low rowing and calf raise at the end of row (with elastic band in standing position); (9) single arm alternating low rowing and calf raise at the end of row (with elastic band in standing position).

According to the initial assessment and response to treatment, medium-level sequences (sequences 4, 5 and 6, shown in Figure 4, Figure 5 and Figure 6) could be introduced.

Figure 4, Sequence 4: (1) side squat from a chair, hands on table; (2) chair side squat; (3) isometric single leg wall squat; (4) side squat, hands on table; (5) side squat; (6) walking side squat; (7) candelabrum side squat; (8) side squat with single front arm elastic band rowing; (9) side squat alternating arm elastic band rowing.

Figure 5, Sequence 5: (1) isometric wall push-up (hands against the wall); (2) dynamic wall push-up; (3) isometric chair push-up (hands on chair seat); (4) chair push-up (dynamic movement, hands on chair seat); (5) easy push-up (knee on the floor); (6) easy double-time push-up (knee on the floor, isometric contraction between concentric and eccentric arm movement); (7) isometric push-up; (8) push-up; (9) push-up with single hip extension.

Figure 6, Sequence 6: (1) table plank, (isometric, hands on table); (2) easy plank plus alternating arm row; (3) chair plank (isometric, hands on chair seat); (4) chair plank alternating single arm row; (5) plank; (6) reverse plank; (7) side plank; (8) dynamic movement from chair plank to chair “v position” (hip flexed at 90°); (9) plank with single hip extension.

After completing the lower-level sequences, high-level sequences (sequences 7 and 8; see Figure 7 and Figure 8) could be administered, if applicable.

Figure 7, Sequence 7: (1) lunge with front leg movement, hands on wall; (2) lunge plus other hip extension movement; (3) half R.O.M. lunge; (4) isometric full R.O.M. lunge; (5) dynamic full R.O.M. lunge; (6) lunge plus double arm rowing; (7) lunge with shoulder abduction; (8) lunge with twist; (9) from dynamic full R.O.M. lunge to calf raise and return.

Figure 8, Sequence 8: (1) table side plank (isometric, hand on table); (2) easy side plank with rotational reach; (3) chair side plank (isometric, hand on chair seat); (4) chair side plank with rotational reach; (5) easy side plank (side plank dynamic movement, one lower limb extended with foot on the floor, the other lower limb positioned forward, with knee flexed at 90° and movement of the foot on the floor, which pushes and facilitates movement; (6) side plank; (7) side plank with single shoulder elevation; (8) side plank with single frontal shoulder abduction; (9) weighted side plank with single frontal shoulder abduction (extra load, 1.5 L water bottle in the free hand).

Participants underwent two sessions per week for 12 weeks. The sequence exercises were the main part of the training session unit. Before this phase, a standardized warm-up of 8 min, consisting of exercises of joint mobility, marching in place and muscle stretching, was carried out. At the end of the session, a standardized cool-down, including exercises of joint mobility and muscle stretching, was administered. FITT parameters: frequency was twice a week; intensity was moderate (3.4 ± 1.0 in the CR10 Borg scale); time (duration) of each session was around 1 h; type was strength training with body weight and resistance band exercises.

### 2.4. Experimental Procedure

#### 2.4.1. Study Protocol

Our research was a clinical study started in April 2019, when we began to search for the participants. The sample was searched via the publication of a social media flyer explaining the characteristics of the study. Volunteers who met the inclusion and exclusion criteria were accepted randomly in order of submission up to the number required by the study. Altogether, 129 subjects proposed themselves for the study. The number of accepted participants was designed to fulfill the statistical criteria and to stay within the limits of our research group. The first 20 interested persons participated in a preliminary study to understand the variability of the data and assess the sufficient sample size (see Statistical Analysis). Once the sample had been established, all the selected participants performed the above-described tests, according to the methods specified below, and the data were managed as reported in the Statistical Analysis. The 12-week training program described in Section 2.4.3 was then carried out, and at the end of the training program, the tests were repeated to verify the effects of the training. The research ended in late December 2019.

In Figure 9 the flowchart of the study.

#### 2.4.2. Tests

The tests took place on two different days, 72 h apart, in the week before the beginning of the HT program (T0). The ones carried out on the first day were: detection of PPE and VAS, SIT REACH and TUG. Those performed on the second day were: STORK, HAND GRIP and CHAIR TEST. As regards the SIT REACH and the TUG, participants were given two trials and the best result was retained for statistical analysis; as for the HAND GRIP, the best result out of three attempts was recorded. All measurements were repeated in the same way in the week following the end of the training program (T1).

#### 2.4.3. Training Progression

The main part of training consisted of 6 total work sets for each session in the first three weeks and at least 8 sets starting from the fourth week. From the third week onward, the sets administered could vary from 8 to 12 according to the individual’s condition, complying with the following progression criteria.

Each subject started with the 3 easiest sequences: 1, 2 and 3. Exercises for lower limbs, trunk and upper limbs were then administered. On the first day of training, all participants underwent exercise 1 of the sequence. They were given two sets of 10 repetitions to perform with the optimal technique. When the subject was able to perform, with the correct technique, 2 sets of 15 repetitions for two consecutive workouts, 1 more set was added, and then he/she could move on to the more complex exercise in the same sequence. If the subject was already able to perform 2 sets of 15 repetitions on the first day, he/she could progress to the upper-level exercise within the same sequence. After 3 weeks, a fourth sequence targeting core stability was added (sequence 6 for all subjects). All data of the training sessions were recorded in the diary registration, as well as any adjustments applied to the exercise program: changes in exercise level and training volume.

#### 2.4.4. Statistical Analysis

Ordinal (perceived physical exertion, PPE), categorical (gender) and continuous (all other) variables were analyzed in the study. Data were collected using Excel 2019 software (Microsoft, Redmond, WA, USA) and processed with SPSS 19 software (IBM Inc, Armonk, NY, USA), and reported as mean and standard deviation. Considering the size of the sample (less than 100), percentage values were reported without decimal place. Normality of data was studied using normality plots, Kolmogorov–Smirnov and Shapiro–Wilk tests. Levene’s test was used to assess homogeneity of variance. Pearson’s and Spearman’s tests were used for the analysis of correlations between variables. The T test was employed to examine the mean differences between pre (T0) and post (T1) training. Cohen’s d effect size was used to investigate the effect size, according to the formula M1-M2/SD pooled (Cohen, 1992), where M1 is the mean value of the first measurement, M2 the mean value of the second test and SD the standard deviation. In some cases, the association between variables was analyzed using the coefficient of determination, CdD (CdD = (r) 2 × 100; r is the value of the correlation coefficient). The significance value was set at 0.05. A preliminary study was conducted to better identify the sample size and organize the exercise criteria. Sample size and statistical power were assessed as described by Cohen (α = 0.5; power value = 0.80) after the pilot study.

## 3. Results

Of the 69 subjects recruited, two interrupted the program because of pathological conditions not due to training (stroke cerebri: 1; fall and femoral fracture during a trip: 1).

Of the 67 patients who completed the study, 15 (22.4%) had a fracture (upper limb: 4; lower limb: 7; spine: 4); 7 (10.4%) a total joint replacement (hip: 3; knee: 4); 24 (35.8%) osteoarthritis (hip: 5; knee: 12; shoulder: 3; other joint: 5); 5 (7.5%) severe spine dysmorphism (scoliosis: 2; thoracic hyperkyphosis: 3); 12 (17.9%) disc herniation (lumbar: 7; thoracic: 1; cervical: 4); no orthopedic pathology was present.

After the application of the HT method, the evaluation of feasibility provided excellent data on adherence: 72% of the participants completed the program and 96% of sessions were completed. Moreover, 97% of the participants displayed progression in the exercise program, with only eight adverse events and, among these, only one interruption of the exercise.

With regard to the aim of verifying the modifications induced by the training program, statistical significance (*p* = 0.001) was found for all the proposed tests.

The main characteristics of the subjects are described in Table 1.

The results of the feasibility analysis of home training are shown in Table 2.

The results of the measurements carried out before and after the training period are shown in Table 3.

On average, the training volume of the main part of the session was 10.5 ± 1.6 sets (lower bound: 7; upper bound: 16).

The improvement in the VAS shows a statistically significant association with the improvement in HAND GRIP (r = 0.45; *p* = 0.001), TUG (r = 0.36; *p* = 0.03) and STORK (r = 0.44; *p* = 0.001).

## 4. Discussion

A new home-based exercise method named home training (HT) was used for the purposes of this study. The effects of its application on a sample of the elderly population in terms of feasibility, physical performance improvement and well-being were investigated. HT is a partially supported method using standardized exercise sequences. The obtained data indicate that HT has good feasibility (adherence, tolerability, safety) and improves strength, functional abilities and well-being.

Among the various possibilities of exercise-based treatment, home-based training has been demonstrated [27] to be a cost-effective method. Other authors [38] have also highlighted that home-based exercise is more effective when associated with a tailor-made program and coaching intervention. The goal of achieving a good therapeutic result in terms of cost-effectiveness using a feasible method was pursued by proposing partially supported home-based exercise. According to this modality, the therapist was always present in the first 3 weeks of the program (6 HT sessions), while, in the remaining 9 weeks (18 HT sessions), he supported only half of the sessions, alternating one session characterized by the therapist’s coaching intervention with a session in which the same program was carried out autonomously by the subject. Thus, 15 out of 24 total sessions scheduled in our study were supported by the therapist. By this approach, it was possible to reach a high level of adherence (see further section, “Feasibility”) and a cost reduction of 38% in therapist costs. Therefore, the cost-effectiveness is determined both by home-based training that avoids the use of specific areas for rehabilitation or training and the fact that the therapist is not present in all sessions.

Past and recent studies have also confirmed the effectiveness of exercise training interventions on physical function in elderly people [19,20,25]. In home-based exercise in general [1], and even more so when choosing a home-based group exercise program for elderly people [39], the prescription of the program has to be even more detailed. The choice of exercises and progression criteria are very important for both the research [1] and adherence to the program [12,21], as well as for its safety and efficacy [6,28]. A home setting requires only minimal equipment. We therefore chose 72 exercises mainly involving the patient’s body weight and, in some cases, elastic bands. The exercises in a sequence ranged from the easiest at level 1 to the most complex at level 9: in case of improvement, according to the progression criteria already described, the patient moved on to the next exercise within the same sequence and not skipping any exercise. Therefore, always starting from the first exercise of the sequence, the therapist chooses, among the exercises, the one that best fits the patient’s condition. In choosing the subject’s ability level within the range of possibilities allowed by the sequence, a form of evaluation of the subject must be carried out, but further validation studies are necessary in this respect. The use of exercise sequences allows the subject to have constant feedback on his/her condition of physical ability. Work progression is characterized not only by the increase in training volume but also by the transition to a higher exercise level—that is, a different and more complex exercise requiring superior motor skills. In this way, the therapist can obtain better subject compliance in supported and unsupported sessions. However, training progression can take place only in supported sessions.

### 4.1. Feasibility

Chen B. [40] imputed the not fully positive effects of home-based exercise on patients’ poor compliance in terms of adherence to the treatment. Mahmood [9] suggested that adherence to home-based treatment should be measured and the term “adherent” should be used for subjects performing more than 70% of the prescribed treatments, while the term “non-adherent” should be associated with those performing a smaller percentage. In our study, 65 participants (97% of the sample) were “adherent”. This is an extremely positive result, which seems to confirm the subjects’ good compliance. As shown in Table 2, 96% of the prescribed treatments were completed by our sample and 65 out of 67 subjects (97%) had sequence progression, which can be explained by the good adherence level measured in both supported and non-supported sessions. In terms of feasibility, the occurrence of adverse events should also be taken into account [12]. The HT sessions completed were 1543 out of a maximum possible amount of 1608, and the number of adverse events (Table 2) occurred can be considered infrequent (eight in total). All physical tests performed at T0 and T1, as well as all of the exercises, were safe and well tolerated. Safety is confirmed by the lack of accidents during the sessions, while the good tolerability of the treatment is demonstrated by the low number of consequences of adverse events that occurred during sessions (eight adverse events) and by the fact that it only once occurred that a session had to be interrupted.

### 4.2. Strength, Functional Ability and Well-Being Effects

The CHAIR TEST and HAND GRIP provide measures of muscle strength, while functional abilities are described by TUG, STORK and SIT REACH. VAS and PPE are to be considered as a general health index, suitable to assess the level of well-being.

The CHAIR TEST mainly describes lower limb muscle strength, which is very important for the prevention of falls and the reduction of frailty in elderly people and in the case of other pathological conditions [1,24,26,40]. HT proved to be effective (31% improvement after training) in improving the CHAIR TEST results. The effect size (large effect size) and significance observed when comparing before and after training measures confirm that HT can be used to improve lower limb strength.

HAND GRIP is a marker of overall body muscle strength and is used to identify frailty and physical activity levels [41]. Data relative to our sample are similar to those reported in the international literature on subjects from Southern Europe [41], but the improvement obtained between T0 and T1 was trivial (effect size: 0.2; difference: 6%) although statistically significant. The HAND GRIP result is basically influenced by four variables (gender, age, general health and nutritional status). The poor improvement in our sample may be due to the fact that, in the short period of treatment with HT, these variables did not change and the sequences mainly involved the lower limbs and trunk muscles.

The TUG has also been used in other studies to assess functional status before and after home-based exercise [42]. In our sample, we found a statistically significant difference between T0 and T1, with a 17% decrease in the time needed to complete the test and a medium effect size, thus confirming that HT improves older people’s functional level. Improvements after 3 months of home-based exercise reported in 2018 by Tsekoura [42] were greater than those observed in our sample; however, the program adopted in the previous research included additional walking training, which may explain the differences reported in the two studies.

In 2017, Jeeyoung H. [43] used the SIT REACH to evaluate functional status in elderly people, reporting statistically significant improvements after 12 weeks of home-based exercise. Although our training program did not contain specific stretching exercises for the hamstrings and lower back muscles, HT similarly produced a marked improvement in the test.

Balance is of the utmost importance for preventing falls, especially in elderly and frail people [44]. The stork balance test has been used in young athletes as well as in elderly and pathological subjects [45]. In all of the exercise sequences, especially those for the trunk and lower limbs, great attention was paid to safely shifting the subject’s center of mass during strength training exercise. As regards the STORK, our data show an effective improvement in balance (65%) between the periods before and after training sessions (large effect size and very good level of significant difference).

Perceived Physical Exertion in daily activity can be influenced by mental exertion, daily hassles and physical exertion or illness and can be measured using the CR10 Borg scale [35]. In our data, PPE in activities of daily living and related well-being status showed a great improvement with HT (mega effect size). Lower fatigue levels related to activities of daily living are very important to maintain adequate rates of physical activity in older people, as, due to age, they might experience a natural decrease in efficiency in activities of daily living [46].

The VAS was used to determine total daily pain [47] or pain during activities of daily living [48]. A medium value of VAS (mean value 5.3 ± 2.3) was observed in our patients before HT. The effect of training with HT on pain during activities of daily living proved to be very good (large effect size and 34% decrease in the average value of VAS). The decrease in VAS values is associated with the improvement measured in the HAND GRIP, TUG and STORK, and the coefficient of determination helps to explain the association between these variables. Only 20% of the HAND GRIP, 13% of the TUG and 19% of the STORK can be associated with pain reduction. Pain therefore seems to have only partial effects on strength, physical performance and balance improvement. The positive effect of HT can be explained, as already indicated by other authors [1,6,17,19,20,21,25,40,44], by the training effects counteracting the negative consequences of aging.

A strong point of the study is its large sample size and the 12-week-long application of the new method, during which period it proved to be economical and to have good feasibility and efficacy. Its limitations are the localization of the sample within a specific geographical area and the difference in number between male (*n* = 25) and female samples (*n* = 44). Future research should employ a sample from a larger geographical area and with a number of male participants nearer to that of the females. Furthermore, the results obtained with HT should be compared with those obtained with other training methods.

## 5. Conclusions

A goal attainment process was used to verify the effects in terms of feasibility, strength and functional abilities of a new method (standardized choice of exercises and progression) designed for home-based exercise. The goals were cost-effectiveness, safety and tolerability and effects on strength, functional abilities and well-being. Strict exercise sequences seem to help to better define the progression criteria in terms of increasing exercise difficulty and controlling the training volume, thus making HT safe and well tolerated. By using the standardized sequences, it was possible to reduce the therapist’s visits and therefore achieve a better cost-effectiveness ratio. Feasibility in terms of adherence and progression was very good. Safety was confirmed by the total absence of accidents during the sessions. All of the outcomes of functional, strength and well-being variables improved after the training, showing that HT, although using moderate training volumes, induced an improvement in all the parameters investigated.

## Figures and Tables

**Figure 1 geriatrics-07-00134-f001:**
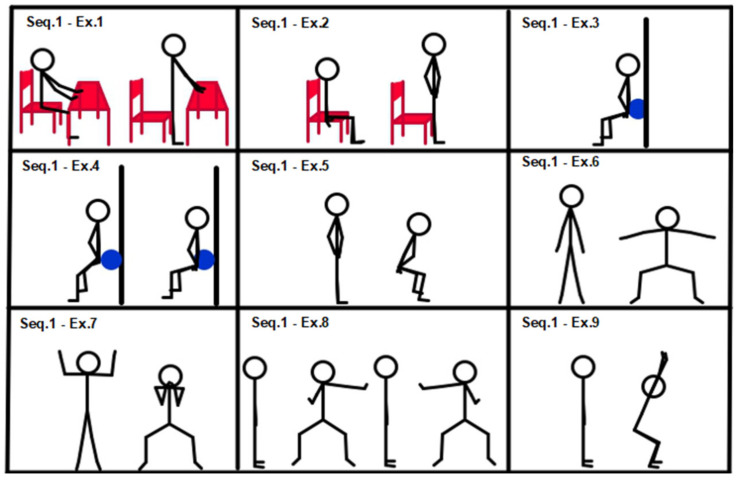
Sequence 1: lower limbs, entry level.

**Figure 2 geriatrics-07-00134-f002:**
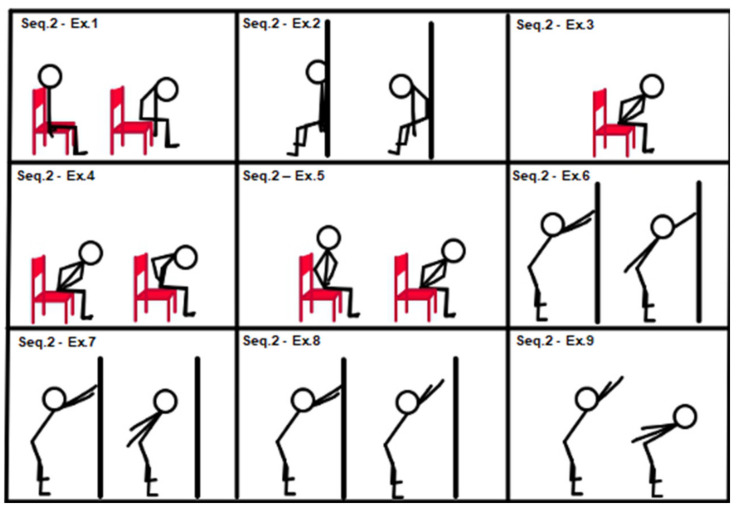
Sequence 2: trunk, entry level.

**Figure 3 geriatrics-07-00134-f003:**
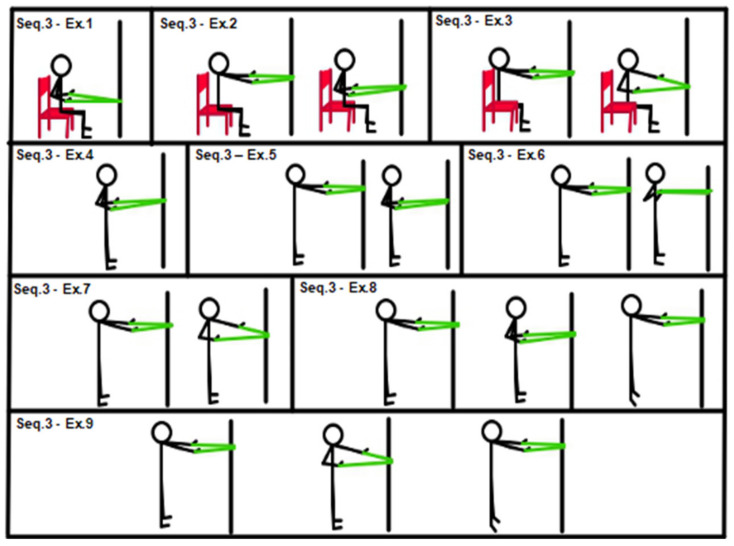
Sequence 3: upper limbs, entry level.

**Figure 4 geriatrics-07-00134-f004:**
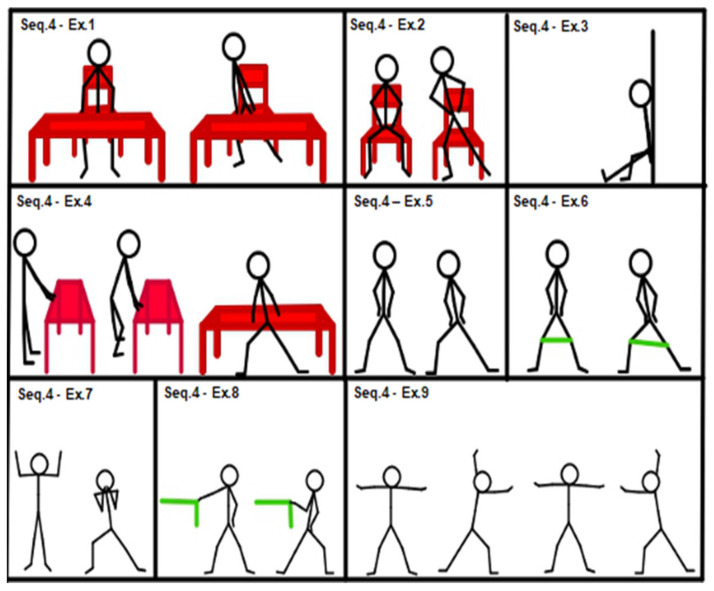
Sequence 4: lower limbs, medium level.

**Figure 5 geriatrics-07-00134-f005:**
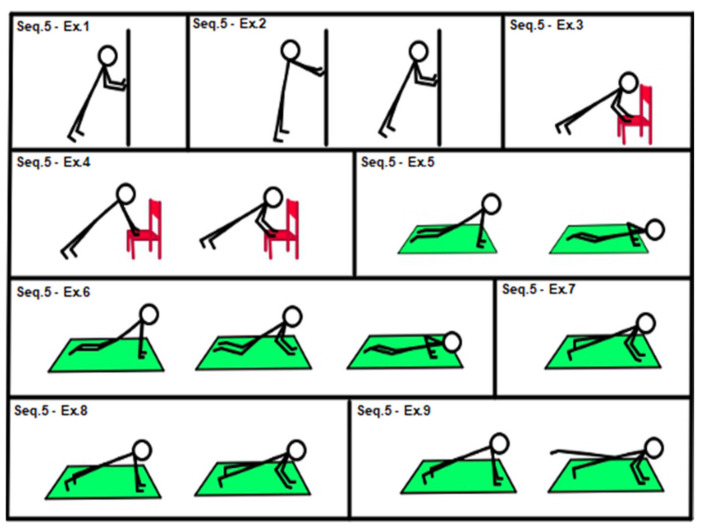
Sequence: 5 upper limbs, medium level.

**Figure 6 geriatrics-07-00134-f006:**
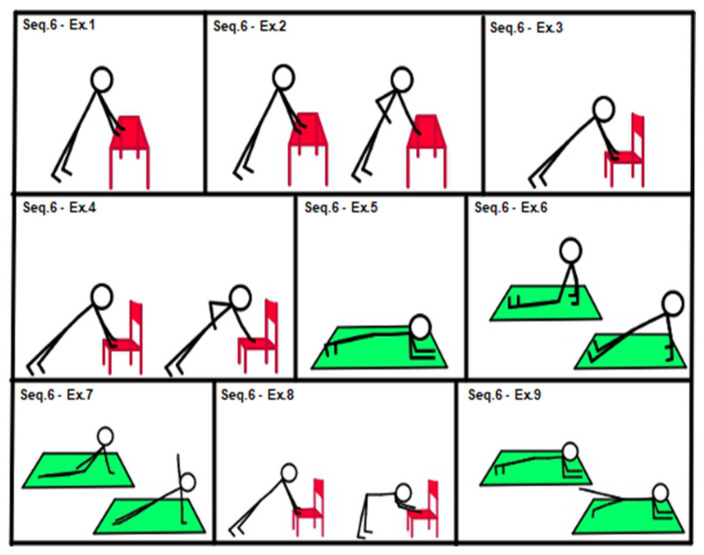
Sequence 6: core stability, medium level.

**Figure 7 geriatrics-07-00134-f007:**
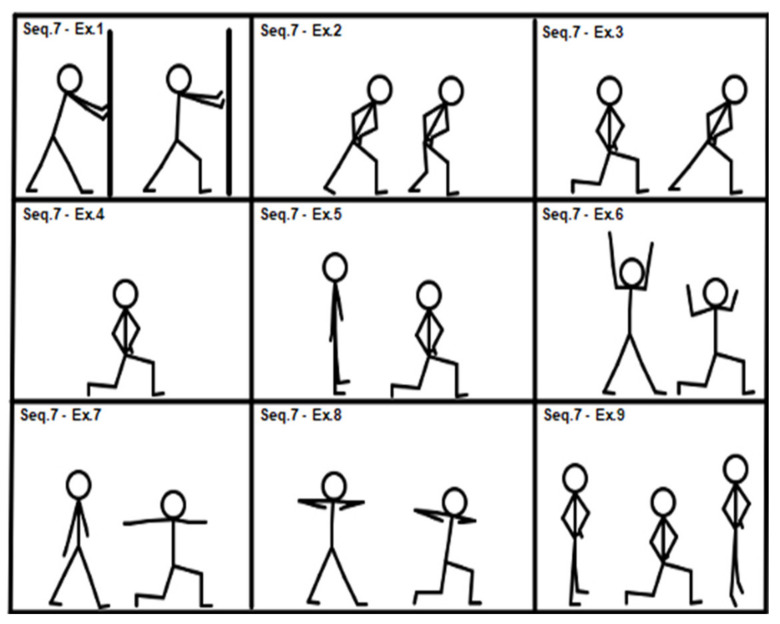
Sequence 7: lower limbs, high level.

**Figure 8 geriatrics-07-00134-f008:**
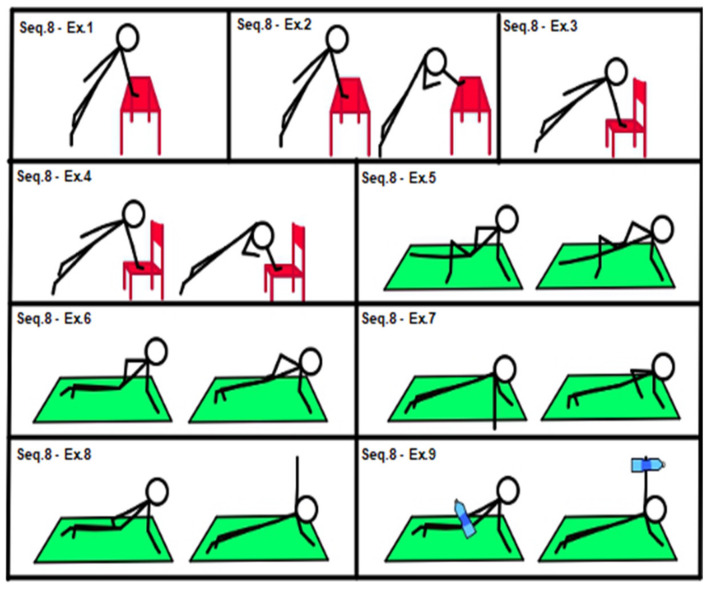
Sequence 8: core stability, high level.

**Figure 9 geriatrics-07-00134-f009:**
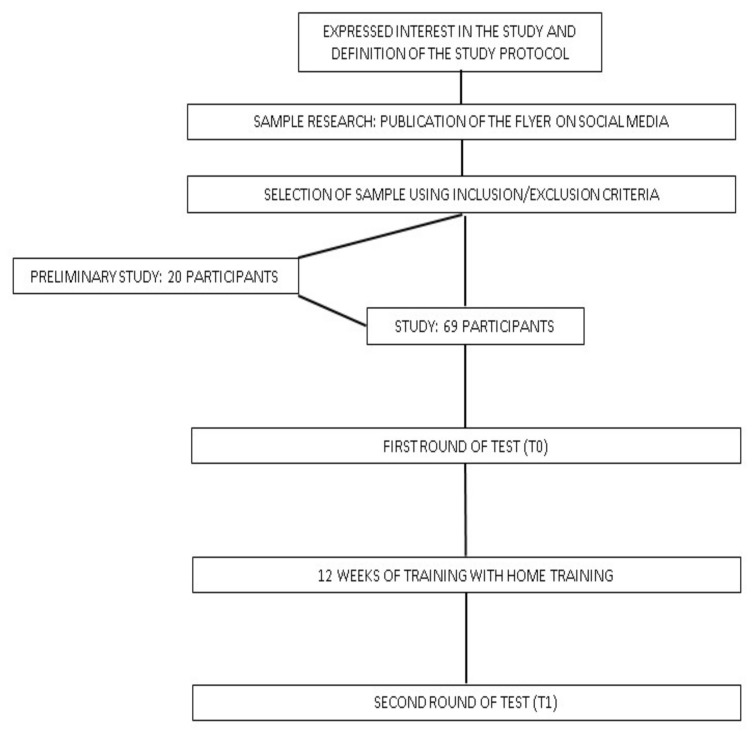
Flowchart of the study.

**Table 1 geriatrics-07-00134-t001:** Description of subjects’ characteristics.

TOT *n* = 69	FEMALE *n* = 44	MALE *n* = 25	MEAN
NATIONALITY	ITALIAN	ITALIAN	
EMPLOYMENT STATUS	RETIRED	RETIRED	-----
AGE (years)	71.8 ± 0.9	74.5 ± 1.2	72.5 ± 5.1
HEIGHT (cm)	159.1 ± 7.0	169.3 ± 8.3	163.3 ± 11.5
BODY MASS (kg)	68.9 ± 12.2	78.8 ± 11.5	73.6 ± 13.1
BMI	27.3 ± 5.0	27.5 ± 3.8	27.3 ± 4.4

**Table 2 geriatrics-07-00134-t002:** Home training feasibility. I.G. is the initial group number.

	I.G. (*n* 67)	Comments
Adherence to HT exercise session		
Patients who completed (*n* (%))	48 (72%)	Private reasons, slightly seasonal illnesses
Sessions completed (*n* (%))	1543 (96%)	
Adjustments of the exercise program		
Progression of exercise program (*n* (%))	65 (97%)	
Regression of exercise program (*n* (%))	0	
No progression or regression (*n* (%))	0	
Both progression and regression (*n* (%))	2 (3%)	
Diary registration (*n* (%))		
All weeks	59 (88%)	
Some weeks	8 (12%)	
No weeks	0	
Adverse events (*n*)	8	Low back pain: 4; knee pain: 3; shoulder pain: 1
Consequences of the adverse events		
None	7	
Discontinuation of the supervised exercise session (*n*)	1	

**Table 3 geriatrics-07-00134-t003:** Average values, standard deviation (sd), effect size and significance value (*p* value) of the measured data before starting the training period (T0) and at the end of the training period (T1). Percentage is the percentage change in T0 value. * = statistically significant.

	CHAIR TEST (reps)		HAND GRIP (kg)		TUG (sec)		SIT and REACH (cm)		STORK TEST (sec)		PPE (A.U. 0–10)		VAS (A.U. 1–10)	
	T0	T1	T0	T1	T0	T1	T0	T1	T0	T1	T0	T1	T0	T1
mean	12.7	16.6	21.2	22.4	7.7	6.4	−6.9	−3.1	20.7	34.2	4.2	1.3	5.3	3.5
sd	3.7	4.6	6.8	7.8	2.3	1.7	2.5	1.8	9.1	11.3	1.1	0.5	2.3	2.4
effect size	−0.9		−0,2		0.7		−1.8		−1.3		3.4		0.8	
percentage	31		6		−17		55		65		−69		−34	
*p* value	0.001 *		0.001 *		0.001 *		0.001 *		0.001 *		0.001 *		0.001 *	

## Data Availability

All study data are included in the present manuscript.

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
