# Peer review of "A Study about a New Standardized Method of Home-Based Exercise in Elderly People Aged 65 and Older to Improve Motor Abilities and Well-Being: Feasibility, Functional Abilities and Strength Improvements"

_geriatrics, 2022, doi:10.3390/geriatrics7060134_

Round 1
Reviewer 1 Report
Conclusion: major revision
The paper focused on exercise for older people home-based, which represents a suitable alternative to conventional center-based cardiac rehabilitation. The present article provides scientific support for this new approach by using a bibliometric analysis to explore important points and frontiers of research in this field. However, the comments below need to be answered before the manuscript can be considered for acceptance.
Introduction:
-Make clearer the aim of this study
-Add previous scientific knowledge on this area
Methods
-Line 69: Please include Mean+_SD
-Line 75. delete etc
-Rewrite inclusion and exclusion criteria.
-Make a better structure in methods including: Research design, sample, data collection, assessments, outcomes, statistical analysis
-Moreover please add a flowchart
-Has the protocol been registered in RCT gov?
- Please define the way of samples’ selection. Was it random? Please describe the way of choosing them. How the sample size was defined?
-Please say more about exercise programme. Describe FITT parameters.
Results
-Needs to be rewritten. Please include statistical results.
Discussion
This section is inadequately described. Below are essential points that should be considered and included for better insight and to increase the impact of the work.
-Please improve your discussion including new evidence
-Please include strengths of the study, study limitations, future research.
-Strength, functional ability and well-being effects:
Statement: Past studies have confirmed the effectiveness of exercise training intervention on physical fuction in elderly. Please provide a citation for this important statement
Effects of multicomponent exercise training intervention on
hemodynamic and physical function in older residents of long-term
care facilities: A multicenter randomized clinical controlled trial. 2021. Journal of Bodywork and Movement Therapies 28, 231-237 https://doi.org/10.1016/j.jbmt.2021.07.009
Pepera G, Krinta K, Mpea C, Antoniou V, Peristeropoulos A, & Dimitriadis Z.
(2022). Randomized Controlled Trial of Group Exercise Intervention for Fall Risk Factors Reduction in Nursing Home Residents. Canadian Journal on Aging / La Revue canadienne du vieillissement. https://doi.org/10.1017/S0714980822000265
Bernocchi, P., Giordano, A., Pintavalle, G., Galli, T., Ballini Spoglia, E., Baratti,
D., et al. (2019). Feasibility and clinical efficacy of a multidisciplinary hometelehealth program to prevent falls in older adults: A randomized controlled trial. Journal of the American Medical Directors Association, 20(3), 340–346. http://doi.org/10.1016/j.jamda.2018.09.003
Miller, K.L., Magel, J.R., Hayes, J.G., 2010. The effects of a home-based exercise program on balance confidence, balance performance, and gait in debilitated, ambulatory community-dwelling older adults: a pilot study. J. Geriatr. Phys. Ther. 33, 85e91.
Author Response
Reviewer 1
Conclusion: major revision
The paper focused on exercise for older people home-based, which represents a suitable alternative to conventional center-based cardiac rehabilitation. The present article provides scientific support for this new approach by using a bibliometric analysis to explore important points and frontiers of research in this field. However, the comments below need to be answered before the manuscript can be considered for acceptance.
Introduction:
- Make clearer the aim of this study
Accepted: the aims have been clarified.
- Add previous scientific knowledge on this area
Accepted: four new citations added to the introduction.
Methods
- Line 69: Please include Mean+SD
Accepted: Mean and sd included
- Line 75. delete etc
Accepted:Deleted
- Rewrite inclusion and exclusion criteria.
Accepted: Inclusion and exclusion criteria rewrited
- Make a better structure in methods including: Research design, sample, data collection, assessments, outcomes, statistical analysis
Accepted: A new paragraph (Study protocol) has been added
- Moreover please add a flowchart
Accepted: Figure 9 (flowchart) has been added
- Has the protocol been registered in RCT gov?
Answer: Researches based on physical exercise in our country do not required to be registered in RCT gov.
- Please define the way of samples’ selection. Was it random? Please describe the way of choosing them. How the sample size was defined?
Accepted: See the new paragraph “Study protocol”.
- Please say more about exercise programme. Describe FITT parameters.
Accepted: FITT parameters have been described.
Results
- Needs to be rewritten. Please include statistical results.
Accepted: Statistical results has been included.
Discussion
This section is inadequately described. Below are essential points that should be considered and included for better insight and to increase the impact of the work.
- Please improve your discussion including new evidence
Accepted: New evidence included
- Please include strengths of the study, study limitations, future research.
Accepted: A part regarding strengths of the study, study limitations, future research has been added in Discussion.
-Strength, functional ability and well-being effects:
14) Statement: Past studies have confirmed the effectiveness of exercise training intervention on physical fuction in elderly. Please provide a citation for this important statement
Accepted: A citation has been added.
Reviewer 2 Report
In the current context of an ageing population, this study is of great interest in the field of prevention. There is, however, a low average age and therefore a population which can participate in external activities. If this type of programme is offered at home, it should be possible to detect the first signs of frailty.
That’s why I think the chosen balance test is not the best. The STORK is a fitness test without clinical value I would have preferred a test like Frail best test.
Apart from that the article is well constructed and respects the rules of publication.
The exercise program is well described and can be a valuable aid for practitioners
The Frail'BESTest. An Adaptation of the "Balance Evaluation System Test" for Frail Older Adults. Description, Internal Consistency and Inter-Rater Reliability. Clin Interv Aging. 2020 Jul 30;15:1249-1262
Author Response
Reviewer 2
Comments and Suggestions for Authors
In the current context of an ageing population, this study is of great interest in the field of prevention. There is, however, a low average age and therefore a population which can participate in external activities. If this type of programme is offered at home, it should be possible to detect the first signs of frailty.
That’s why I think the chosen balance test is not the best. The STORK is a fitness test without clinical value I would have preferred a test like Frail best test.
Answer: We’ll consider this suggestion for future research.
Apart from that the article is well constructed and respects the rules of publication.
The exercise program is well described and can be a valuable aid for practitioners
Round 2
Reviewer 1 Report
The authors replies to all comments. I endorse this paper to get published.